# A Novel Approach for the Identification of Pharmacogenetic Variants in *MT-RNR1* through Next-Generation Sequencing Off-Target Data

**DOI:** 10.3390/jcm9072082

**Published:** 2020-07-02

**Authors:** Javier Lanillos, María Santos, Marta Carcajona, Juan María Roldan-Romero, Angel M. Martinez, Bruna Calsina, María Monteagudo, Luis Javier Leandro-García, Cristina Montero-Conde, Alberto Cascón, Paolo Maietta, Sara Alvarez, Mercedes Robledo, Cristina Rodriguez-Antona

**Affiliations:** 1Hereditary Endocrine Cancer Group, Human Cancer Genetics Programme, Spanish National Cancer Research Centre (CNIO), 28029 Madrid, Spain; jlanillos@cnio.es (J.L.); msantosr@cnio.es (M.S.); jmroldan@cnio.es (J.M.R.-R.); ammontes@cnio.es (A.M.M.); bcalsina@cnio.es (B.C.); mmonteagudo@cnio.es (M.M.); ljleandro@cnio.es (L.J.L.-G.); cmontero@cnio.es (C.M.-C.); acascon@cnio.es (A.C.); mrobledo@cnio.es (M.R.); 2Nimgenetics, 28049 Madrid, Spain; mcarcajona@nimgenetics.com (M.C.); pmaietta@nimgenetics.com (P.M.); salvarez@nimgenetics.com (S.A.); 3Centro de Investigación Biomédica en Red de Enfermedades Raras (CIBERER), 28029 Madrid, Spain

**Keywords:** pharmacogenomics, hearing loss, aminoglycoside antibiotics, off-target next-generation sequencing (NGS) data, *MT-RNR1*, bioinformatics

## Abstract

Specific genetic variants in the mitochondrially encoded 12S ribosomal RNA gene (*MT-RNR1*) cause aminoglycoside-induced irreversible hearing loss. Mitochondrial DNA is usually not included in targeted sequencing experiments; however, off-target data may deliver this information. Here, we extract *MT-RNR1* genetic variation, including the most relevant ototoxicity variant m.1555A>G, using the off-target reads of 473 research samples, sequenced through a capture-based, custom-targeted panel and whole exome sequencing (WES), and of 1245 diagnostic samples with clinical WES. Sanger sequencing and fluorescence-based genotyping were used for genotype validation. There was a correlation between off-target reads and mitochondrial coverage (r_customPanel_ = 0.39, *p* = 2 × 10^−13^ and rWES = 0.67, *p* = 7 × 10^−21^). The median read depth of *MT-RNR1* m.1555 was similar to the average mitochondrial genome coverage, with saliva and blood samples giving comparable results. The genotypes from 415 samples, including three m.1555G carriers, were concordant with fluorescence-based genotyping data. In clinical WES, median *MT-RNR1* coverage was 56×, with 90% of samples having ≥20 reads at m.1555 position, and one m.1494T and three m.1555G carriers were identified with no evidence for heteroplasmy. Altogether, this study shows that obtaining *MT-RNR1* genotypes through off-target reads is an efficient strategy that can impulse preemptive pharmacogenetic screening of this mitochondrial gene.

## 1. Introduction

Mitochondria are cellular organelles specialized in energy production with varying abundance across tissues and cell types, depending on the metabolic demand. Mitochondria have their own genetic material as single or multiple copies of a circular double-stranded DNA that is 17 kb long, which is maternally inherited and encodes the information for 37 genes. One of them is *MT-RNR1*, which codes for the 12S ribosomal subunit [1]. Specific variants in this gene increase the binding affinity of aminoglycosides to the mitochondrial ribosome and are highly susceptible to irreversible hearing loss after administration of aminoglycoside antibiotics, regardless of dose, length of treatment, or serum drug levels. The most frequent cause for this ototoxicity is the variant m.1555A>G. This variant has been associated with hearing loss after treatment with aminoglycosides (e.g., gentamicin, streptomycin, tobramycin, kanamycin) in pedigree studies and in individual cases in multiple populations, with nearly 100% of variant carriers who receive aminoglycoside developing hearing loss [2,3,4,5].

In addition to m.1555A>G, other *MT-RNR1* variants associated with this ototoxicity have been described [2,3]. Together with m.1555A>G, the associations of m.1494C>T and m.1095T>C mtDNA variants with aminoglycoside-induced hearing loss are well-supported within the literature, and together with m.1555A>G, are included in international pharmacogenetics databases (e.g., PharmGKB). The number of *MT-RNR1* variants with pharmacogenetic implications has been growing based on the increasing number of studies focused on patients with diverse clinical conditions who required treatment with aminoglycosides [6], showing a large spectrum of *MT-RNR1* pharmacogenetic variation [2,3,5,6,7,8,9,10,11,12].

Advances in NGS technologies have drastically increased the amount of genomic sequence data. The impact and immersion of NGS technologies in the clinics has been illustrated by national and international genomic medicine initiatives all over the globe [13,14,15], boosting the access to targeted and preemptive pharmacogenomics. With the advent of NGS data and the determinant role of the mtDNA variation in some diseases, the development of bioinformatics to explore mtDNA variation has been prolific these past years [16,17], and continues evolving [18,19]. There is a plethora of bioinformatics tools for robust detection (e.g., filtering “nuclear mitochondrial DNA” segments, also known as NUMTs) and annotation of pathogenic mtDNA variation, characterization of mtDNA haplogroups, as well as various mtDNA-specific databases [16]. However, the development of mtDNA variation databases is slow, and genomic aggregation initiatives (e.g., gnomAD), which provide accurate mutation prevalence across different populations, have not yet reported mtDNA variation [20]. Furthermore, despite the strong immersion of NGS in the clinics [13,14,15], the clinical implementation of *MT-RNR1* genetic testing is not extensive [21].

The mtDNA genetic information used in previous pharmacogenetic studies linked to aminoglycoside-induced ototoxicity has been preferentially obtained by low-throughput strategies based on Sanger sequencing, mtDNA-directed microarray, or polymerase chain reaction (PCR) specific amplification of mtDNA, followed by restriction enzyme digestion and capillary electrophoresis [3,4,12,22,23]. Nowadays, whole genome sequencing (WGS) includes whole mtDNA data at high depth of coverage [24], but it is an expensive technique that requires significant computational and storage data analysis. Hybridization capture-based targeted NGS technology, such as whole exome sequencing (WES), enable more rapid, affordable, and multiplexed analysis of the genetic regions of interest. However, most commercial NGS library preparation kits for targeted capture experiments are designed to target a fraction of the nuclear genome and evade mtDNA screening, hindering preemptive genetic testing for these variants. Accidental off-target capture of mtDNA fragments during hybridization in NGS experiments has been found useful to explore mtDNA variation in cancer genomics research and detect pathogenic mtDNA variants in mitochondrial genetic disorders [25,26,27,28]. However, it is unknown whether this could be a suitable approach for mtDNA pharmacogenetics, and more specifically, whether it could constitute a novel effective approach to determine the *MT-RNR1* genotype.

In this study, we used 473 research DNA samples, sequenced with a targeted custom panel or WES, and 1245 diagnostic samples that had undergone clinical WES to compare off-target data and MT coverage, extract *MT-RNR1* reads, and infer m.1555A>G genotypes. These genotypes derived off-target were validated with an alternative method. Overall, we show that this novel approach for *MT-RNR1* genotype determination is effective and constitutes an opportunity to boost mtDNA pharmacogenetics using NGS technologies.

## 2. Experimental Section

### 2.1. Samples

We analyzed 473 germline DNA samples (403 blood, 38 saliva, 23 frozen normal kidney tissue, and nine formalin-fixed paraffin-embedded (FFPE) normal kidney tissue) belonging to 443 unrelated cancer patients, most from Spanish origin (95%) and used for other purposes in research projects [11,12]. In 21 individuals, gene panel-targeted data were available from both blood and saliva. The other nine individuals had both WES and gene-targeted panel data from blood. DNA from 456 samples were available for genotyping/Sanger sequencing.

We also included in the study 1245 unrelated individuals, not previously described, that had undergone clinical WES for diagnosis of potential hereditary diseases. From these cases, DNA samples and clinical data were not available.

Research samples and their NGS data were dissociated from personal information, and the NGS data derived from the clinical samples were anonymized to avoid any personal identification. The project was approved by the IRB at Instituto de Salud Carlos III (PI 47_2020-v2, 11 June 2020).

### 2.2. Targeted Capture Next-Generation Sequencing

For 325 research samples, NGS data corresponded to a custom gene panel (SeqCap EZ Choice Enrichment Kit, Roche, Basel, Switzerland) targeting ~135 kb of the coding region of 29 cancer-related genes. DNA libraries were prepared following the manufacturers’ guidelines in four groups of ≤96-plexed samples using 250–500 ng of input DNA and with some technical modifications in the library preparation procedure among the groups (e.g., double-size selection optimization in Groups 2 and 3). Each group of samples was sequenced separately in a HiSeq 2500 (Illumina) in 100 × 2 mode.

For 148 research samples, NGS data corresponded to Whole Exome Sequencing (WES) prepared using SureSelectXT or XT-HS and Human All Exon capture probes (Agilent), following the manufacturer’s guidelines and sequenced on a HiSeq or Novaseq (Illumina).

The 1245 clinical samples were subjected to WES aimed at diagnosis of hereditary conditions using SureselectXT (Agilent) and a Human All Exon v6 kit (Agilent, Santa Clara, CA, USA), and sequenced in a NovaSeq6000.

A summary of samples used, as well as library preparation kits and sequencing runs for research samples and clinical samples are presented in Appendix A, respectively.

### 2.3. Bioinformatics Analysis

From the 473 samples obtained from research projects, paired-end sequencing raw data (Fastq files) were retrieved, and all were analyzed under the same NGS-processing pipeline using the corresponding BED file. In brief, our in-house pipeline was used as input FASTQ files in both read-pair directions, trimmed from adapter contamination. Next, the tool BWA-mem (v0.7.17-r1188, Wellcome Trust Sanger Institute, Cambridge, UK) aligned the reads to the GRChr37 (humanG1Kv37, GATK4 v4.0.5.1, (The Broad Institute of Harvard and MIT, Cambridge, MA, USA) human genome reference, which included the mitochondrial chromosome (MT:1-16569). Aligned reads (BAM) were sorted and deduplicated using Samtools (v1.9, Wellcome Trust Sanger Institute, Cambridge, UK; The Broad Institute of Harvard and MIT, Cambridge, MA, USA) and Picard (v2.18.7, The Broad Institute of Harvard and MIT, Cambridge, MA, USA), respectively. Sequencing QC metrics were extracted with Picard (CollectHsMetrics). These Picard HsMetrics output files were parsed and merged into a unique metrics file for subsequent analysis. Specific mtDNA off-target data were extracted from the initial BAM into new mtDNA BAM files for the next analysis. The read counting of the mtDNA off-target sequencing reads mapped to the mitochondrial chromosome (chrMT:1-16569) from the resulting BAM files was performed using bam-readcount (v0.8.0-unstable-6-963acab-dirty, bam-readcount, Github) with default parameters. Each sample generated a CSV table with read count information for all mtDNA positions and different nucleotides. A Python script was developed to parse and merge all sample CSV files together and perform subsequent analysis. Off-target reads at m.1555 were extracted and samples were classified as wild type or carriers. In research samples, we assumed homoplasmy due to their low mtDNA coverage. 

From the 1245 samples with clinical WES, the mtDNA off-target data (BAM files) were generated from NGS raw data (paired-end FASTQ files) by performing a two-step alignment. To reduce computation times, raw data were aligned only to the mtDNA chromosome. The resulting reads mapped to the mtDNA chromosome were converted back to FASTQ and realigned to the whole human reference genome (including the mtDNA chromosome) to relocate back to the nuclear genome those reads with high similarity to the mtDNA (i.e., nuclear mtDNA sequences, NUMTs). Read counting and metrics calculation of the mtDNA data available in the final BAM files was performed as described above. WES coverage estimations were calculated with GATK DepthofCoverage.

For the *MT-RNR1* genotype assignment using off-target NGS data, variant calling was performed with Mutect2 (GATK4, v4.1.6.0) in each BAM file of 1245 clinical WES samples containing the mtDNA off-target reads. Sample VCF files were merged into a single multi-sample VCF with Bcftools (v1.9, Wellcome Trust Sanger Institute, Cambridge, UK). Variant annotation was performed using the Variant Effect Predictor (VEP v94.4, European Molecular Biology Laboratory, Cambridge, UK) and VCF was parsed into a table with a custom Python script. *MT-RNR1* variants with 10% Variant Allele Fraction (VAF) or higher were considered.

### 2.4. Genotyping and Sanger Sequencing of m.1555A>G

All DNA samples available in the study (*n* = 456, corresponding to 443 unrelated individuals) were genotyped for m.1555A>G using the KASP SNP Genotyping System (LGC Biosearch Technologies, Novato, CA, USA), a fluorescence (FRET)-based assay using competitive allele-specific PCR, according to the vendor’s instructions. Samples were genotyped in duplicates. Samples were labeled as “undetermined” when both duplicates were uninformative.

All samples identified as m.1555G carriers were also analyzed by Sanger sequencing using primers covering chrMT:1489–1675 positions (5′-CCGTCACCCTCCTCAAGTAT-3′ and 5′-TAACTTGACCGCTCTGAGCT -3′) and designed to specifically amplify mtDNA and avoid amplification of similar sequences in the nuclear genome.

We performed a BLASTn search of the sequence spanning 300 bases downstream and upstream of the m.1555 variant to identify highly homologous regions in nuclear chromosomes (e.g., chromosome 5, 7, 9, and 11). For the fluorescence-based genotyping assay, we indicated the best regions to select specific primers, and for Sanger sequencing we manually chose primers with mismatches at the 3′ end in the homologous nuclear regions.

### 2.5. Statistical Methods

The percentage of off-target reads mapping to the mtDNA chromosome in each sample was calculated as the fraction between the number of reads mapping to the mtDNA and the number of reads mapping off-target regions. Linear regression was used to assess the association between off-target reads and mtDNA coverage using the “stats.linregress” function from Scipy (Python 3.6). Histogram plots were generated with the “distplot” function (“step” mode) from the Seaborn library (Python 3.6). 

### 2.6. Data Availability

BAM files containing mitochondrial mapped off-target sequencing reads of all samples described in the article are available at the ENA EMBL-EBI repository (PRJEB38987).

## 3. Results

### 3.1. Off-Target Sequencing Metrics

NGS data from 473 germline DNA research samples, mainly blood (85%, *n* = 403) and saliva (8%, *n* = 38), but also normal kidney tissue (7%, *n* = 32) corresponding to 443 unrelated individuals were analyzed to obtain off-targets. DNA libraries of the research samples corresponded to a custom gene panel (*n* = 325) or to WES (*n* = 148) (Table 1). The median sequencing depth for the custom gene panel samples was 2M read-pairs and for WES samples 44M read-pairs. The corresponding capture designs yielded a median on-target coverage of 890× in the custom gene panel and 86× in WES.

The off-target was higher in the custom gene panel samples than in WES (50% versus 15%), which is an expected inverse correlation between this metric and the design-specific size (135 KB and ~60 MB, respectively) in NGS hybrid capture-based target enrichment experiments. A small fraction of the off-target reads were mapped to the mtDNA (0.03% and 0.02%), resulting in 6× and 21× median mtDNA genome coverage, in the custom gene panel and WES samples, respectively (Table 1). The mtDNA coverage is higher in WES samples since they are sequenced at higher sequencing depths. We found a positive correlation between the proportion of off-target reads and the mtDNA coverage (r_WES_ = 0.67, *p* = 7 × 10^−21^ and r_customPanel_ = 0.39, *p* = 2 × 10^−13^; Figure 1). In addition to the panel design size, the off-target varied with specific controlled library preparation conditions in each experiment. This variation in off-targets per experiment is shown by the four groups of custom gene panel samples (Figure 1C). 

Regarding the type of sample, saliva showed two-fold higher mtDNA coverage when compared to blood (11× versus 5×). FFPE normal kidney samples showed the highest off-target and mtDNA coverage (custom gene panel: 73% and 93×; WES: 18% and 40×, respectively; Table 1), which in comparison with frozen kidney samples (WES: 5% and 5×), suggests that samples with decreased DNA integrity results in better mtDNA coverage. 

### 3.2. MT-RNR1 m.1555A>G Detection Using Off-Target Data

When reads at m.1555A>G were extracted, 35% (*n* = 91) of the custom gene panel and 62% (*n* = 113) of WES research samples had five or more reads at this position (Figure 2). The median coverage at m.1555 in the custom gene panel was 4×, 14×, and 96× for blood (*n* = 283), saliva (*n* = 38), and the FFPE normal kidney (*n* = 4), respectively. In WES, the coverage was higher, with 20×, 40×, and 2× for blood (*n* = 120), FFPE normal kidney (*n* = 5), and frozen normal kidney (*n* = 23), respectively. Histograms showing m.1555 coverage distribution for blood and saliva samples are presented in Figure 2.

Regarding m.1555A>G genotype classification, 431 research samples were classified as wild type, and three as m.1555G carriers. The latter were supported by only one, two, and three G-reads (none for the reference), but this variant was homoplasmic in most cases. To validate the results, a *MT-RNR1* m.1555A>G fluorescence-based genotyping assay was carried out. Among 456 DNA samples available, 447 were classified as wild-type, and four as m.1555G carriers—the three samples previously detected, plus one sample with no off-target reads at m.1555 (Figure 3A). The genotype of 24 individuals with two samples was concordant. In total, in 415 samples with both genotyping and NGS off-target data, there was full concordance in the genotype assignation (Figure 3B). The genotypes of the four m.1555G carriers were also validated through Sanger sequencing (Figure 3C).

### 3.3. Clinical WES Samples for Off-Target MT-RNR1 Genotype Assignation

We extracted the mtDNA off-target data derived from clinical WES of 1245 unrelated individuals. The on-target median coverage was 125× (additional details in Appendix A). The median coverage across the mtDNA chromosome was 59× and the coverage of *MT-RNR1* gene was 56×. The WES of clinical and research samples displayed a similar mtDNA coverage pattern, being higher in clinical WES, while the custom panel samples had a lower mtDNA coverage and displayed a different profile (Figure 4). The average sequencing raw data generated for research WES samples and clinical WES samples was 7.9 GB and 15.0 GB, respectively, suggesting this was the main cause driving the differences in their median mtDNA coverage obtained using the off-target reads.

At the m.1555 position, 90% of clinical samples had ≥20× and the median coverage was 53×. From the initial 1245 clinical WES, 1242 individuals were classified as wild type and three as m.1555G carriers (with 26, 54, and 66 reads at this position, all supporting 1555G). Six samples with >40× coverage had one single m.1555G read and the remaining reads supported an A, and were classified as wild type, representing either artifacts or very low heteroplasmy. Thus, 0.24% of the clinical samples (3 of 1245) of individuals were m.1555G carriers.

Appendix A provides the full list of *MT-RNR1* variants detected in the clinical samples, including pathogenic variants associated with aminoglycoside-induced hearing loss, variants of unknown significance, and polymorphisms with no clinical impact. For other relevant variants, such as m.1494C>T and m.1095T>C, the median coverage in the corresponding positions was 54× and 47×, respectively, and one single individual carrying m.1494T was identified (35×: 34 T reads; 1 C read).

Table 2, in addition to m1555A>G and m.1494C>T, provides *MT-RNR1* variants of still unclear clinical significance that were detected in our study and previously suggested to be related with aminoglycoside-induced ototoxicity. We found that 247 samples (20%) had variants in these new 14 selected positions, with six samples having two different variants, and the rest presenting only one. The variant m.827A>G was the most common within these samples (43%), followed by m.827A>G and m.1189T>C. Variants m.961T>C and m.961T>G were detected in nine individuals.

Based on the adequate mtDNA coverage of our clinical WES samples, we searched for variants with a Variant Allele Fraction (VAF) within 0.1 and 0.9, suggestive of heteroplasmy. In seven samples, evidences of heteroplasmy were found (m.827A>G, m.930G>A, m.1189T>C, and m.1462G>A variants).

## 4. Discussion

Mitochondrial DNA is not usually included in targeted sequencing experiments, and currently, large genomic variation databases lack this information. It has already been shown that off-target sequencing data, after proper bioinformatic processing, can be used to determine alterations in the mtDNA genome, providing not only mitochondrial variant frequency across populations but also extending the diagnostic potential of NGS. However, it is currently unknown whether this approach could be suitable for *MT-RNR1* genotype determination [30,31]. In this study, by using a large dataset of research and clinical samples from different targeted NGS experimental strategies (custom gene panel and WES), we demonstrate the feasibility of using off-target reads to inform for clinically relevant *MT-RNR1* pharmacogenetic variants.

Using off-target NGS reads for mitochondrial DNA sequencing has already been well-described in mtDNA genetic disorders and cancer genomics [25,26,27,28]. A key issue consists of efficiently filtering nuclear mitochondrial DNA segments (NUMTs) derived reads and retrieving true mtDNA-mapped reads [32]. Several mtDNA genomic studies have applied diverse tools and coverage, read mapping, base quality, and variant allele fraction filtering criteria to detect true mtDNA variation and avoid low complexity and NUMT regions [25]. Here, we leveraged the information of the mitochondrial *MT-RNR1* gene, which permitted the inspection of research samples of diverse sources and the inference of the *MT-RNR1* m.1555A>G genotype. After validation of the genotype results by alternative genotyping methods, we analyzed a large set of clinical samples analyzed by WES, most of which (90%) showed ≥20 read depth at position m.1555. This off-target mitochondrial coverage has previously been shown to reliably call for genetic variants with different heteroplasmy levels, with a rate of rare sequencing errors similar to that observed with conventional Sanger sequencing [25,32]. Heteroplasmy levels at the m.1555A>G variant has been previously shown to modify the hearing loss penetrance [4,33]. Thus, genetic testing of this variant should aim at detecting mtDNA heteroplasmy. It has been suggested that heteroplasmy in 0.1–0.9 levels can be reliably detected using off-target reads, based on variant allele fraction and total read depths, with a 30×-fold read depth [26], which occurs in 82% of our clinical WES samples. At any rate, all seven m.1555G carriers identified in this study were consistent with homoplasmic variants according to the different techniques used for genotype assignation. Regarding the type of samples, blood and saliva are the most commonly used sources of germline DNA for NGS testing [34,35]. In our study, they show similar performance, suggesting that various germline DNA sample types can be used for off-target NGS genetic testing, in contrast with WGS [34].

The association between *MT-RNR1* variants and antibiotic-induced hearing loss has been investigated in a significant number of familial or disease-focused studies [7]; however, the relatively small number of individuals studied, and the low frequency of the pathogenic variants does not allow for the accurate determination of the population frequencies. Estimated m.1555A>G carrier frequencies are within the 0.2–0.3% range [36,37], while mtDNA genetic databases offer estimates of ~0.6% [38], depending on the population. Overall, we found 0.4% prevalence (1:240; 0.92% in research samples and 0.24% in the high-quality clinical samples) in our study with mainly Spanish individuals. For *MT-RNR1* m.1494C>T and 1095T>C, also associated with nonsyndromic hearing loss in the presence of aminoglycoside exposure [2], but less common than m.1555A>G, only one carrier individual was identified.

Characterization of mtDNA genetic variation using unintendedly produced off-target sequencing data can impulse the advancement of preemptive pharmacogenetic screening and the prevention of aminoglycoside-induced hearing loss, for example, in diagnostic WES in children with suspected genetic diseases [7,15]. In addition, our work provides the possibility of further exploring the relevance of this marker using available off-target NGS data in diverse clinical fields, including patients with tuberculosis, with cystic fibrosis, and surgical patients allergic to beta-lactam antibiotics or oncology (e.g., patients with febrile neutropenia caused by chemotherapy) [2,5,8,9,10,11,12,23,29].

In total, 20% of the clinical samples with WES carried at least one *MT-RNR1* variant previously reported in connection with aminoglycoside-induced hearing loss (Table 2). However, further studies are needed to validate the clinical meaning of many of these variants, which until then, should be considered variants of unknown significance. In addition, bioinformatic curation of these variants is required [17,18]. For example, the reference variant m.961T is flanked by five cytosine bases downstream and four cytosine bases upstream, and this must be taken into account when calling indels in homopolymeric regions [17,19,39,40]. Thus, a combination of NGS and clinical data will help in the characterization of *MT-RNR1* variation, boosting its clinical interpretation. In this regard, off-target mtDNA data derived from NGS is a useful source of information.

The caveats of using off-target sequencing data to attain mtDNA genetics testing will be overcome with the inclusion of the mtDNA sequence in targeted capture designs. In addition, the increased availability of WGS data from national and international genomic medicine initiatives [14,41], together with the growing interest in retrieving mtDNA variation from these thousands of individuals sequenced with WGS [24], will aid in the implementation of mitochondrial pharmacogenomic variants. Nevertheless, there is still a vast amount of targeted capture NGS data from WES [20] and custom gene panels that can be explored retrospectively and prospectively to provide deeper knowledge of mtDNA pharmacogenetic variations and boost the clinical implementation of *MT-RNR1* testing. In conclusion, this study proves that obtaining *MT-RNR1* genotypes from off-target sequencing reads is an efficient approach that could be used for clinical testing.

## Figures and Tables

**Figure 1 jcm-09-02082-f001:**
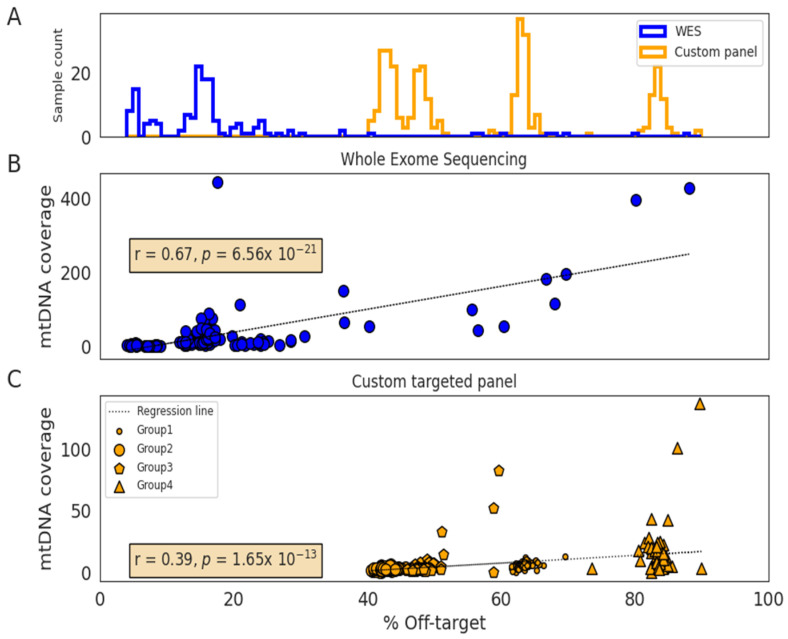
Distribution of off-targets according to the sequencing method and correlation with mitochondrial DNA coverage. (**A**) Histogram showing the off-target distribution of whole-exome sequencing (WES) (*n* = 148; blue) and custom panel (*n* = 325; orange) samples. (**B**) Correlation between off-target and mtDNA coverage in WES samples (*n* = 148). (**C**) Correlation between off-target and mtDNA coverage in the custom-targeted gene panel samples (*n* = 325). These samples were sequenced into four different library preparation next-generation sequencing (NGS) experiments (Group 1–4).

**Figure 2 jcm-09-02082-f002:**
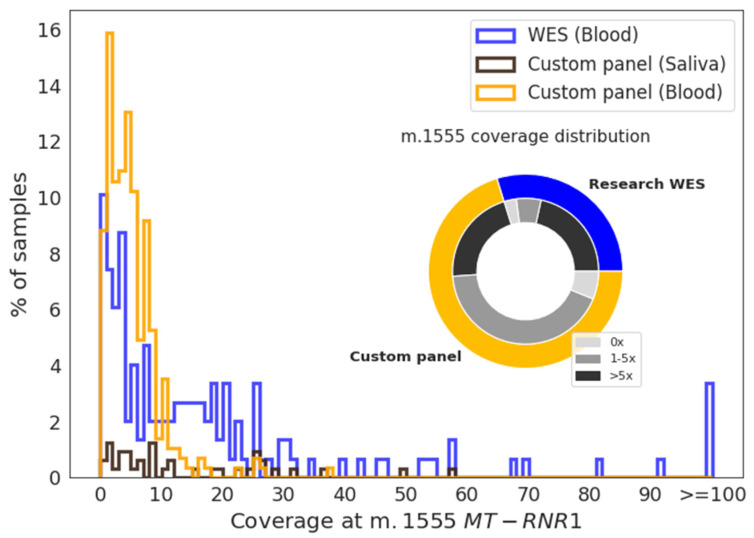
Coverage distribution at the m.1555 position. Histogram showing the coverage of blood samples analyzed by whole-exome sequencing (WES) (blue) or a custom gene panel (orange) and of saliva samples (dark brown) at m.1555. The pie chart shows the proportion of blood samples within each group with a coverage of 0×, 1–5×, or >5× at m.1555.

**Figure 3 jcm-09-02082-f003:**
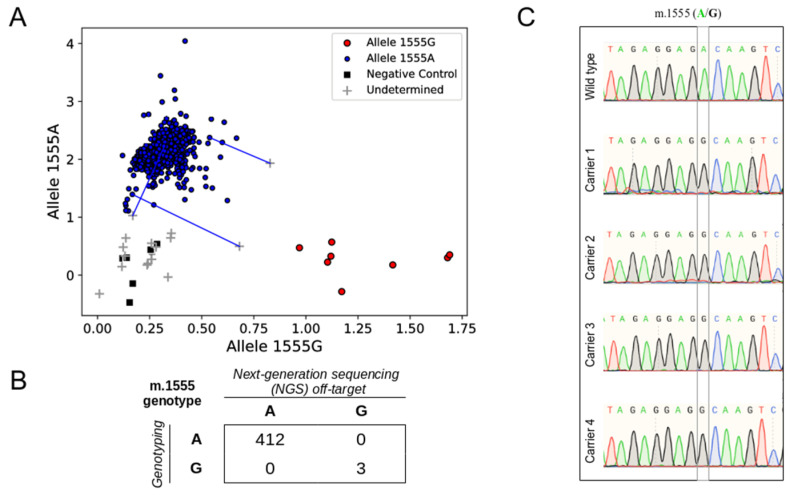
Validation of *MT-RNR1* m.1555A>G genotypes identified by off-target reads. (**A**) Genotype clusters generated by a fluorescence (FRET)-based assay. Wild-type samples are colored in blue, samples carrying m.1555G are colored in red, non-template controls are shown as black squares, and uninformative samples are shown with a gray cross. Samples were genotyped in duplicates. Blue lines connect uninformative outliers with their wild-type duplicates. (**B**) Distribution of the *MT-RNR1* m.1555A>G genotypes obtained with next-generation sequencing (NGS) off-target and fluorescence-based genotyping. (**C**) Sanger sequencing chromatograms corresponding to a wild-type sample control (Wild type) and the four m.1555G carriers identified among the research samples.

**Figure 4 jcm-09-02082-f004:**
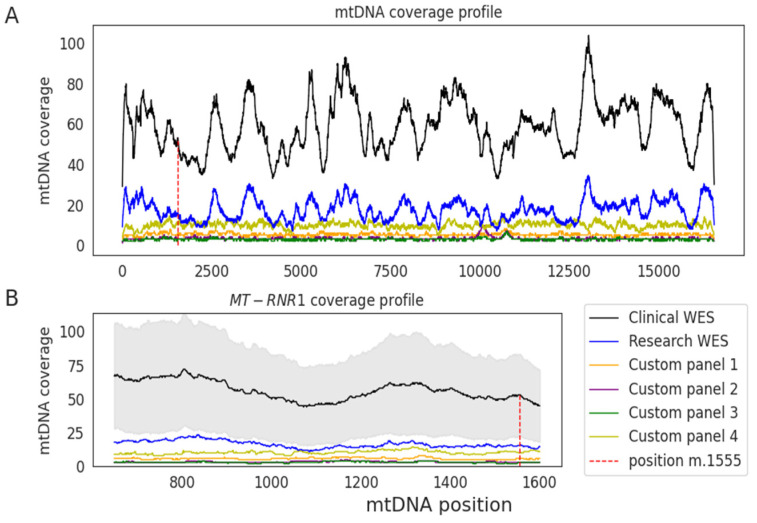
Median read depth across the mitochondrial genome using off-target reads. (**A**) Median coverage is shown for clinical whole-exome sequencing (WES) (black) and for research samples analyzed through WES (blue) or a custom-targeted panel (different colors indicate different sample sets) across each position of the mitochondrial genome. (**B**) Median coverage in the *MT-RNR1* gene. Shaded area shows the standard deviation of the clinical WES samples.

**Table 1 jcm-09-02082-t001:** Summary of samples and coverage for mtDNA and the m.1555 position.

Sample Type	Number of Samples	Median Coverage	
		Mitochondria	m.1555
**Research samples-custom panel**			
Blood	283	5	4
Saliva	38	11	14
Kidney FFPE	4	93	96
**Research samples-WES**			
Blood	120	21	20
Kidney frozen	23	5	2
Kidney FFPE	5	18	40
**Clinical samples-WES**			
Blood	1245	59	53

**Table 2 jcm-09-02082-t002:** *MT-RNR1* variants related with aminoglycoside ototoxicity found in 1245 clinical whole-exome sequencing (WES) samples.

CHROM ^1^	POS ^2^	REF ^3^	ALT ^4^	N. ^5^	Heteroplasmy ^6^, Sample (VAF ^7^)	Association with Aminoglycoside-Induced Ototoxicity	(References)
chrM	1555	A	G	3	-	Strong	[2,3,4,5]
chrM	1494	C	T	1	-		[2]
chrM	669	T	C	2	-	Further studies needed	[5]
chrM	827	A	G	107	MT384 (0.71); MT1358 (0.12)		[8,29]
chrM	896	A	G	1	-		
chrM	930	G	A	19	MT1043 ^8^ (0.93)		
chrM	961	T	C	7	-		
chrM	961	T	G	2	-		
chrM	988	G	A	1	-		[5]
chrM	1005	T	C	1	-		[9,29]
chrM	1048	C	T	6	-		[9]
chrM	1189	T	C	54	MT1370 (0.86); MT368 (0.86); MT392 (0.89)		[10]
chrM	1243	T	C	11	-		[11]
chrM	1438	G	A	26	-		[12]
chrM	1462	G	A	8	MT652 (0.719)		
chrM	1537	C	T	4	-		[23]

^1^ CHROM: chromosome. ^2^ POS: genomic position. ^3^ REF: reference nucleotide. ^4^ ALT: alternative nucleotide. ^5^ N.: The number of samples carrying the variant. ^6^ Heteroplasmy: variants with 0.10-0.90 Variant Fraction (VAF) and more than five reads supporting the reference and alternative variant. ^7^ VAF: Variant Allele fraction/frequency, that is, the level of heteroplasmy calculated as the fraction of alternative reads from the total number of reads at the variant position. ^8^ The m.930G>A variant in MT1043 was considered further after manual detection (despite VAF > 0.90) since its coverage was 168×.

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
