# Peer review of "A Novel Approach for the Identification of Pharmacogenetic Variants in MT-RNR1 through Next-Generation Sequencing Off-Target Data"

_jcm, 2020, doi:10.3390/jcm9072082_

Round 1

Reviewer 1 Report

Dear Authors, 

Thank you for your work in this area. Overall I found this paper to be a pleasure to read and I have only minor comments listed below. 

----

Minor Criticisms

  • Please add a statement about ethical approval (e.g. IRB, etc) for this study. 
  • The authors mention that the data will be available in SRA/EMBL. Please consider updating this prior to publication. 
  • The authors note that they assumed homoplasmy in these research samples (line 107). Would recommend that they provide a brief comment on that decision. 
  • In the methods (line 130), the authors provide information about primer sequences. Please consider adding more information about how these were designed to exclude NUMTs. 
  • In figure 1, the authors mention library preparation methods as the driver for variation in % off target. Could consider commenting on the different sizes of the libraries (WES vs targeted), or the specific differences in library preparation that might drive this difference. 

Author Response

POINT BY POINT RESPONSE

REVIEWER 1

Dear Authors,

Thank you for your work in this area. Overall I found this paper to be a pleasure to read and I have only minor comments listed below.

Response: Thank you for your kind comments and for identifying specific points that will help to improve the quality of our manuscript. Please, find below our responses and changes performed in the manuscript.

Minor Criticisms

Point 1: Please add a statement about ethical approval (e.g. IRB, etc) for this study.

Response: The statement regarding the ethical approval of this project has been added in the text (Experimental section):

“Research samples and their NGS data were dissociated from personal information, the NGS data derived from the clinical samples was anonymized to avoid any personal identification. The project is approved by the IRB at Instituto de Salud Carlos III (PI 47_2020-v2).”

Point 2: The authors mention that the data will be available in SRA/EMBL. Please consider updating this prior to publication.

Response: BAM files of all the samples in this study with the mtDNA reads have been submitted to the EMBL-ENA repository under PRJEB38987 accession number. This accession number has been updated in the manuscript (Experimental section):

“BAM files containing mitochondrial mapped off-target sequencing reads of all samples described in the article is available at ENA EMBL-EBI repository (PRJEB38987).”

Point 3: The authors note that they assumed homoplasmy in these research samples (line 107). Would recommend that they provide a brief comment on that decision.

Response: We assumed homoplasmy in the research samples in order to classify them as m.1555A>G wild type or carrier because they had low mtDNA coverage (Figure 2), which makes heteroplasmy detection unreliable. We have added this brief explanation in the updated manuscript (Experimental section):

“Off-target reads at m.1555 were extracted and samples were classified as wild type or carriers. In research samples we assumed homoplasmy due to their low mtDNA coverage.

Point 4: In the methods (line 130), the authors provide information about primer sequences. Please consider adding more information about how these were designed to exclude NUMTs.

Response: following your suggestion, we have explained in more detail the primer design which maximized the chances of excluding NUMTs (Experimental section):

“We performed a BLASTn (https://blast.ncbi.nlm.nih.gov/) search of the sequence spanning 300 bases downstream and upstream the m.1555 variant to identify highly homologous regions in nuclear chromosomes (e.g. chromosome 5, 7, 9 and 11). For the fluorescence-based genotyping assay, we indicated the best regions to select specific primers, for Sanger sequencing we manually chose primers with mismatches at the 3’ end in the homologous nuclear regions”.

Point 5: In figure 1, the authors mention library preparation methods as the driver for variation in % off target. Could consider commenting on the different sizes of the libraries (WES vs targeted), or the specific differences in library preparation that might drive this difference.

Response: Thanks for these details. We have rephrased the concerning paragraph including a comment addressing the relationship between the off-target and the library preparation size:

- We point out that in NGS experiments a larger panel design leads to lower off target, which is also observed in our study when comparing a custom gene panel (~135 KB) versus WES (~60 MB).

- Also, we have added a comment about the higher sequencing depth in WES samples, which leads to more mtDNA reads, despite the custom panel samples showed higher % off-target.

- Finally, we have made more explicit how off-target was influenced by controlled experimental conditions. This effect was clear with our custom panel samples in Figure 1C. In the Experimental Section, we specify explain what was the main experimental condition (double-size selection) that modified the % off-target in these sets of samples, which enables to optimize the protocol, following vendor’s tips.

Please, find below the three highlighted sentences with the changes proposed above (Results):

“The off-target was higher in the custom gene panel samples than in WES (50% versus 15%), which is an expected inverse correlation between this metric and the  design-specific size (135 KB and ~60 MB, respectively) in NGS hybrid capture-based target enrichment experiments. A small fraction of the off-target reads were mapped to the mtDNA (0.03% and 0.02%), resulting in 6x and 21x median mtDNA genome coverage, in the custom gene panel and WES samples, respectively (Table 1). The mtDNA coverage is higher in WES samples since they are sequenced at higher sequencing depths. We found a positive correlation between the proportion of off-target reads and the mtDNA coverage (rWES=0.67, p=7x10-21 and rcustomPanel=0.39, p=2x10-13; Fig. 1). In addition to the panel design size, the off-target varied with specific controlled library preparation conditions in each experiment. This variation in off-target per experiment is shown by the four groups of custom gene panel samples (Fig. 1C).”

Reviewer 2 Report

In the present study “Identification of pharmacogenetic variants in MT3 RNR1 through next generation sequencing off-target” Javier Lanillos et al. provide the computational analysis of off target reads for 473 DNA samples, sequenced through a capture-based custom targeted panel or whole exome sequencing (WES) and 1245 diagnostic samples that had undergone clinical WES. The aim of the study is to develop a novel computational tool that will be used in the MT-RNR1 genotype determination.

 In order to improve the present study all the following points should be fully addressed.

  • Lane (67-69) “We analysed 473 germline DNA samples (403 blood, 38 saliva, 23 frozen normal kidney tissue and 9 formalin-fixed paraffin-embedded (FFPE) normal kidney tissue) belonging to 443 unrelated cancer patients”

Why the authors decided to analyse cancer-patient samples? There is any link between the variants in MT3 RNR1 and cancer? Please, explain it in the text.

  • Lane (73-75) We also included in the study 1245 unrelated individuals, not previously described, that had undergone clinical WES for diagnosis of potential hereditary diseases

Why the authors decided to use 1245 sample that had undergone clinical WES for diagnosis of potential hereditary diseases? Please, can the authors explain this aspect in the text?

Aminoglycoside antibiotics cause irreversible hearing loss, and variant 1555A>G in MT-RNR1 causes multiorgan mitochondrial disorder. I think for the present study will make much more sense use samples from patients treated with aminoglycoside antibiotics or patients that shows mitochondrial disorders.

  • Lane (79-80) For 325 research samples NGS data corresponded to a custom gene panel (SeqCap EZ Choice Enrichment Kit, Roche) targeting ~135 kb of the coding region of 29 cancer-related genes.

It is not clear why the authors used for this study, NGS data that correspond to cancer-related genes. I think need to be explained in the manuscript

  • (Figure 1B) Correlation between off-target and mtDNA coverage in WES

I would show even the correlation between off-target and mtDNA coverage in the custom gene panel.

  • (Figure 1C) and the four sets of custom gene panel samples.

Which genes are presented in this figure and why they are used in this study? Please explain it in the figure legend

  • (Title) “Identification of pharmacogenetic variants in MT3 RNR1 through next generation sequencing off-target”

I would change the title that does not give useful information regarding the present study. I suggest change the title and highlights the novelty of this study, that seems to focus on a novel approach for MT-RNR1 genotype determination.

Author Response

POINT BY POINT RESPONSE

REVIEWER 2

In the present study “Identification of pharmacogenetic variants in MT3 RNR1 through next generation sequencing off-target” Javier Lanillos et al. provide the computational analysis of off target reads for 473 DNA samples, sequenced through a capture-based custom targeted panel or whole exome sequencing (WES) and 1245 diagnostic samples that had undergone clinical WES. The aim of the study is to develop a novel computational tool that will be used in the MT-RNR1 genotype determination.

In order to improve the present study all the following points should be fully addressed.

Response: Thank you for kindly accepting to review of our manuscript and identifying key questions that have helped to improve our manuscript.

We acknowledge that there is a common and important issue across your comments related with the general description and use of the individuals/samples included in our study (cancer patients and hereditary diseases). Our main purpose in this study was to determine the feasibility of extracting very relevant mitochondrial DNA pharmacogenetic variants in MT-RNR1 from accidentally generated off-target NGS data. Our study addresses this challenge using diverse available NGS data. In this initial proof-of-concept study, the patients included are not selected base on ototoxicity, since we only want to prove the suitability/limitations of the method. In future studies we plan to use this strategy in clinical samples for which MT-RNR1 variants will have a direct clinical application. Thus, our data collection was not designed to go beyond the purpose of determining the accuracy of this method. The application of the method to a specific patient population for which MT-RNR1 genotype is relevant, is the most interesting aspect but it will be performed in future projects.

Please, find below our specific responses and changes applied in the text.

Point 1: Lane (67-69) “We analysed 473 germline DNA samples (403 blood, 38 saliva, 23 frozen normal kidney tissue and 9 formalin-fixed paraffin-embedded (FFPE) normal kidney tissue) belonging to 443 unrelated cancer patients”

Why the authors decided to analyse cancer-patient samples? There is any link between the variants in MT3 RNR1 and cancer? Please, explain it in the text.

Response: As explained above, we used cancer-patient samples only based on the availability of NGS data (we provided a general description of the samples in the Experimental Section, indicating that it comes from other previous genomic projects related to cancer).

However, it could be of interest to determine the potential relevance of MT-RNR1 variation in cancer patients treated with aminoglycoside antibiotics, since bacterial infections represent an important morbility and mortality cause in pediatric cancer patients (Barbarino, Bitner-Glindzicz, Jing, Padma). However, this study cannot address this topic, since that specific data (e.g. febril neutropenias, use of aminoglycoside antibiotics, ototoxicity) is not included in our database. At any rate, the mtDNA data from this project is going to be available for any researcher willing to explore further into this idea.

We now better remark how MT-RNR1 variation obtained with our approach could contribute to fields like oncology. Please, find this newly added comment highlighted in the text (Discussion):

”In addition, our work provides with the possibility to further explore the relevance of this marker using available off-target NGS data in diverse clinical fields, including patients with tuberculosis, individuals with cystic fibrosis, surgical patients allergic to beta-lactam antibiotics and oncology (e.g. patients with febrile neutropenia caused by chemotherapy)[2]”

Point 2: Lane (73-75) We also included in the study 1245 unrelated individuals, not previously described, that had undergone clinical WES for diagnosis of potential hereditary diseases

Why the authors decided to use 1245 sample that had undergone clinical WES for diagnosis of potential hereditary diseases? Please, can the authors explain this aspect in the text?

Aminoglycoside antibiotics cause irreversible hearing loss, and variant 1555A>G in MT-RNR1 causes multiorgan mitochondrial disorder. I think for the present study will make much more sense use samples from patients treated with aminoglycoside antibiotics or patients that shows mitochondrial disorders.

Response: The possibility of including 1245 clinical WES represented a good opportunity to increase the evidence regarding the retrieval of mtDNA from off-target NGS data using clinical samples. Although our purpose did not go beyond the collection of a large clinical WES dataset to test this approach, we have emphasized in the that the access to any clinical information regarding these 1245 unrelated individuals is restricted:  “DNA samples and clinical data were not available”.

Our future interest relies on more clinically-focused projects, applying our approach to datasets and project designs which will allow us to address the clinical impact of preemptive mtDNA pharmacogenetics in different fields (oncology, pediatry, etc).

We hope that this idea is well explained and sufficiently highlighted in the Discussion:

”Characterization of mtDNA genetic variation using unintendedly produced off-target sequencing data can impulse the advancement of preemptive pharmacogenetic screening and the prevention of aminoglycoside-induced hearing loss, for example, in diagnostic WES in children with suspected genetic diseases [7,15]”).

This is now reinforced by a newly added comment.

 ”In addition, our work provides with the possibility to further explore the relevance of this marker using available off-target NGS data in diverse clinical fields, including patients with tuberculosis, individuals with cystic fibrosis, surgical patients allergic to beta-lactam antibiotics and oncology (e.g. patients with febrile neutropenia caused by chemotherapy)[2,5,8–12,23,29]”).

Point 3: Lane (79-80) For 325 research samples NGS data corresponded to a custom gene panel (SeqCap EZ Choice Enrichment Kit, Roche) targeting ~135 kb of the coding region of 29 cancer-related genes.

It is not clear why the authors used for this study, NGS data that correspond to cancer-related genes. I think need to be explained in the manuscript.

Response: Thanks for this comment, because it is related to a main message of this work, which is that any laboratory with access to similar NGS datasets would have this new opportunity to recover mtDNA data from the off-target data, regardless of the gene panel they used. Please, see also our response to Point 1.

In the case of those 325 samples, their libraries were designed to capture 29 genes related to cancer for specific oncology projects. We have not provided more information regarding the genes captured because the fraction of data used in this project is not coming from the targeted genes, but the off-target that was generated during the library preparation.

We hope to better explain this point by adding complementary explanations across the manuscript:

- We introduced a comment to clarify that the actual data presented in this manuscript was extracted from the initial data generated, so that no information about the genes targeted is included, but only mtDNA data derived from off-target reads:

“Specific mtDNA off-target data was extracted from the initial BAM into new mtDNA BAM files for the next analysis.”

- In the Discussion, we recall the use of two different panels in this project, aiming to motivate using NGS data from diverse panels available:

“In this study, by using a large dataset of research and clinical samples from different targeted NGS experimental strategies (custom gene panel and WES), we demonstrate the feasibility of using off-target reads to inform for clinically relevant MT-RNR1 pharmacogenetic variants.”.

Point 4: (Figure 1B) Correlation between off-target and mtDNA coverage in WES

I would show even the correlation between off-target and mtDNA coverage in the custom gene panel.

Response: The correlation between off-target and mtDNA coverage in the custom gene panel (n=325 samples) is shown in Figure 1C (Custom targeted panel). We have modified the legend in Figure 1 to better clarify that the relationship between off-target (x-axis) and mtDNA coverage (y-axis) was displayed separately by sample type (WES or custom gene panel samples). Figure 1B shows only WES samples, and Figure 1C the custom targeted panel samples, but both figures show the same variables for different sets of samples.

Point 5: (Figure 1C) and the four sets of custom gene panel samples.

Which genes are presented in this figure and why they are used in this study? Please explain it in the figure legend

Response: Please see our response to Point 4. Figure 1 shows mtDNA coverage per sample and its relationship with the off-target data. Specifically, Figure 1C displays off-target derived mtDNA data from the 325 custom gene samples, not from any specific gene.

These 325 samples underwent a custom gene panel library preparation, in four different NGS experiments, which are defined as Group 1, 2, 3, and 4 (each sample can be uniquely identified in which NGS experiment was prepared and sequenced by searching in column “SETS” of the Supplementary Table 1). Each dot in Figure 1C represents a sample, and there are 4 different dot shapes (4 sets/groups) in order to visualize to which of the four library preparation experiments they belonged. We have introduced a clarification regarding these four groups, which may help to justify the visualization:

In addition to the panel design size, the off-target varied with specific controlled library preparation conditions in each experiment. This variation in off-target per experiment is shown by the four groups of custom gene panel samples (Fig. 1C).

We have modified the legend of Figure 1C to provide a clearer description.

Point 6: (Title) “Identification of pharmacogenetic variants in MT3 RNR1 through next generation sequencing off-target”

I would change the title that does not give useful information regarding the present study. I suggest change the title and highlights the novelty of this study, that seems to focus on a novel approach for MT-RNR1 genotype determination.

Response: We agree the title must help to focus on the purpose of our study, which is the novel approach and the opportunity to extract important pharmacogenetics mitochondrial variants from NGS experiments using off-target data. Hence, we propose the following modification in the title:

A novel approach for the identification of pharmacogenetic variants in MT-RNR1 through next generation sequencing off-target data”

Thank you for this suggestion.

Round 2

Reviewer 2 Report

Dear Authors,

Thank you for your clarifications.

Good luck for the future and looking forward to see the application of the present method to specific patient population for which MT-RNR1 genotype is relevant.

This manuscript is a resubmission of an earlier submission. The following is a list of the peer review reports and author responses from that submission.